

# Cryogenic cave minerals recorded 1889 CE melt event in northeast Greenland

Anika Donner[1], Paul Töchterle[1], Christoph Spötl[1], Irka Hajdas[2], Xianglei Li[3], R. Lawrence Edwards[4], Gina
E. Moseley[1]

[1]Institute of Geology, University of Innsbruck, Austria
[2]Laboratory of Ion Beam Physics, ETH Zurich, Switzerland
[3]Institute of Earth Environment, Chinese Academy of Sciences, China
[4]Department of Earth Sciences, University of Minnesota, USA

*Correspondence to:* Anika Donner (anika.donner@uibk.ac.at)





**Abstract.** The investigation of cryogenic cave minerals (CCMs) has developed in recent decades to be a particularly valuable proxy for palaeo-permafrost reconstruction. Due to difficulties, however, in obtaining reliable chronologies with the so-called "fine" form of these minerals, such studies have thus far utilised the "coarse" form. In this study, we successfully investigate the northernmost-known deposit of fine-grained cryogenic cave minerals (CCMs), which are situated in Cove Cave (Greenlandic

translation: Eqik Qaarusussuaq), a low-elevation permafrost cave in northeast Greenland (80° N). The Cove Cave CCMs display a complex mineralogy that consists of fine-grained cryogenic cave carbonates ($CCC_{fine}$) as well as sulphate minerals (gypsum, eugsterite, mirabilite, and löweite). In comparison to $CCC_{fine}$ from the mid-latitudes, positive $\delta^{13}C$ values (7.0 to 11.4 ‰) recorded in Cove Cave are similar. In contrast, Cove Cave $CCC_{fine}$ $\delta^{18}O$ values are ca. 8 to 16 ‰ lower. Furthermore, despite previous $CCC_{fine}$ dating efforts being unsuccessful, here we demonstrate that precise dating is possible with both isochron-based $^{230}Th/U$

dating and $^{14}C$ dating if the dead carbon fraction is reliably known.

The dating result (65±17 a BP; 1885±17 CE) shows that the CCMs formed during the late Little Ice Age, a time interval characterised by cold temperatures and abundant permafrost in northeast Greenland, making water infiltration into Cove Cave dependent on water amount and latent heat. We relate the CCM formation to a combination of black carbon deposition and anomalously high temperatures, which occurred over a few days, in the summer of 1889 CE. Such extreme conditions led to

widespread melting over large areas of the Greenland ice sheet. We propose that the anomalous (weather) conditions of 1889 CE also affected northeast Greenland, where the enhanced melting of a local ice cap resulted in water entering the cave and rapidly freezing. While $CCC_{fine}$ and gypsum likely precipitated concurrently with freezing, the origin of the other sulphate minerals might not be purely cryogenic but could be linked to subsequent sublimation of this ice accumulation in the very dry cave environment.

**1 Introduction**

In recent decades, cryogenic cave carbonates (CCCs), a type of speleothem associated with the formation of cave ice, have become a valuable tool for tracking evidence of past permafrost presence, particularly in the mid-latitudes in either low-elevation temperate locations (e.g., central Europe; Richter et al., 2018; Žák et al., 2012) or high-elevation periglacial environments (Bartolomé et al., 2015; Luetscher et al., 2013; Spötl et al., 2021; Spötl and Cheng, 2014). In contrast, investigations into CCCs from high-latitude

caves are rare with the exception of a few studies from northern Yukon, Canada (Clark and Lauriol, 1992; Lauriol et al., 1988; Lauriol and Clark, 1993).

The presently accepted mechanism for CCC formation is precipitation from freezing karst water (Žák et al., 2012). $CCC_{coarse}$ are typically found in micro-climatically stable environments of cave interiors, where they precipitate in slowly freezing pools of water carved into cave ice deposits by drip water, making them a useful proxy for the reconstruction of palaeo-permafrost (Žák et al.,

2018). In contrast, $CCC_{fine}$ precipitate from a rapidly freezing film of water on top of cave ice and are commonly found in well ventilated caves and/or near cave entrances and thus may be related to local thermal anomalies rather than regional permafrost (Žák et al., 2012). In addition to CCCs, other minerals are also known to form cryogenically in caves, e.g., gypsum and other sulphate minerals, collectively referred to as cryogenic cave minerals (CCMs; Žák et al., 2018). In this paper, we follow the approach of Žák et al. (2018) and use the term (fine-grained) CCMs to describe our samples, whilst also comparing them to the

CCM subtypes $CCC_{coarse}$ and $CCC_{fine}$ from previous studies.

$CCC_{coarse}$ can be radiometrically dated using standard $^{230}Th/U$ techniques (e.g., Koltai et al., 2021; Luetscher et al., 2013; Spötl et al., 2021; Žák et al., 2012), although the accuracy of $^{230}Th/U$ ages using standard evaluation procedures has been recently called



into question (Töchterle et al., 2022). Dating of $CCC_{fine}$, on the other hand, has proven to be a difficult task. $^{14}C$ dating of $CCC_{fine}$ suffers from large uncertainties associated with estimating the initial radiocarbon activity and the dead carbon fraction (DCF)

(Lauriol and Clark, 1993), whereas $^{230}Th/U$ dating has been hindered by poor age precision due to very high detrital thorium contamination, which is particularly challenging in young (i.e., late Holocene) samples (Spötl, 2008; Spötl and Cheng, 2014). Due to these unsuccessful dating efforts, the potential of $CCC_{fine}$ as a palaeoclimate archive has yet to be realised.

In this study, we investigate fine-grained CCMs from a low-elevation permafrost cave (Cove Cave (unofficial name); Eqik Qaarusussuaq (Greenlandic)) in northeast Greenland, currently the northernmost-known cave containing cryogenic mineral

deposits, which is located within a climatically highly sensitive region adjacent to the Greenland ice sheet (Bintanja and Krikken, 2016; Bintanja and Selten, 2014; Shepherd, 2016). This paper aims to: i) extend the existing knowledge on morphology, mineralogy, and stable isotopic composition of fine-grained high-latitude CCMs; ii) constrain the age of fine-grained CCMs (and $CCC_{fine}$) using $^{230}Th/U$ and $^{14}C$ dating methods; and iii) ascertain the circumstances of CCM formation at this location.

**2 Study Site**

Cove Cave (80.25° N, 21.93° W) is located in a small tributary valley to a steep-sided canyon on Kronprins Christian Land in northeast Greenland (Fig. 1; Moseley et al., 2020). The Silurian limestones and dolostones in this area host numerous solution caves (Smith and Rasmussen, 2020), some of which were first discovered in 1960 (Davies and Krinsley, 1960), followed by more cave discoveries during three subsequent caving expeditions (Loubière, 1987; Moseley et al., 2020). The area is located ca. 35 km

from the coast (eastwards) and ca. 60 km from the margin of the Greenland ice sheet (south-westwards). It is characterised by an arid climate (ca. 200 mm $a^{-1}$; Schuster et al., 2021), permanently frozen ground, sparse soil and vegetation cover, and permafrost landforms (Moseley et al., 2020). Under contemporary climatic conditions, common speleothems (i.e., stalactites, stalagmites, and flowstones) can therefore not form in caves due to the lack of water infiltration. Weather stations or long-term weather observations are absent in the area, however, mean annual air temperatures at the nearest weather station on the ice sheet (KPC_L PROMICE

weather station, 07/2008–12/2021; Fausto et al., 2019; van As et al., 2011) and near the coast (Station Nord, 1991–2020; Danmarks Meteorologiske Institut, 2022) are -13.5°C and -15.1°C, respectively. Compared to these two stations, a more continental climate with warmer summer temperatures is expected for the study site (Donner et al., 2020). In 1960 and 1983, a small ice cap was present on the plateau in close vicinity to Cove Cave but this has since melted (Fig. 1; Davies and Krinsley, 1960; Loubière, 1987; Moseley et al., 2020; Moseley et al., 2021). Elsewhere in the area, local plateaus are now ice free in summer (Moseley et al., 2021),

though there is ample geomorphological evidence that ice caps were present in the past (Fig. 1; Sole et al., 2020), which might have covered the plateau above Cove Cave.



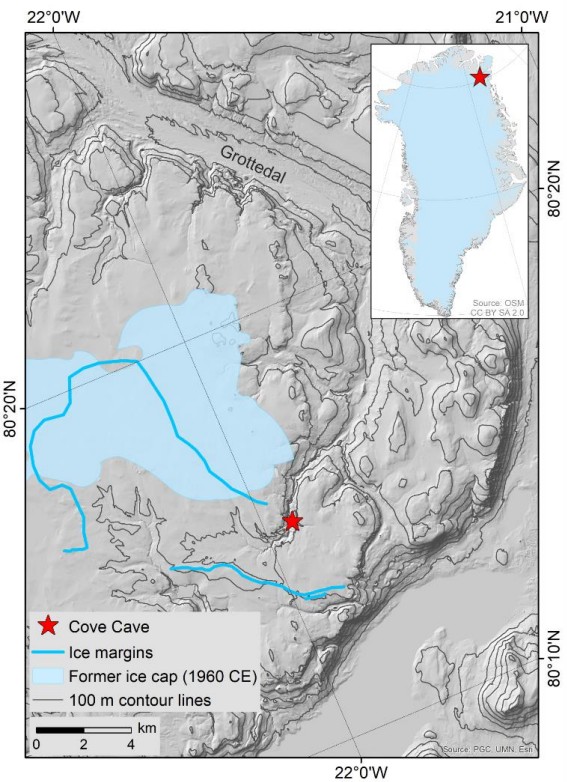

**Figure 1**: Location of Cove Cave in a tributary valley of the larger Grottedal valley in northeast Greenland (map insert: Geofabrik and OpenStreetMap Contributors, 2018). The cave is located close to margins of former ice caps. The most recent ice cap was mapped in 1960
(Davies and Krinsley, 1960), still existed in 1983 (Loubière, 1987) and disappeared before 2019 (Moseley et al., 2020). Geomorphological evidence of ice margins indicate the former presence of additional ice caps, which have not been dated (Sole et al., 2020).

Cove Cave consists of a 103 m long, gently dipping phreatic passage, making it currently the longest explored cave in Greenland (Fig. 2; Moseley et al., 2020). The cave hosts both CCMs and inactive flowstones, the latter indicating at least one phase of warmer and wetter climate in the past. The cave entrance is located 660 m above sea level with a rock overburden of ca. 25 m. Inside the
entrance area, an ice pond, ice stalagmites, and hoar frost have been observed during a visit in summer (Moseley et al., 2020). Beyond the entrance area, the cave is devoid of ice. Flowstone drapes the walls of a vadose canyon and broken angular flowstone blocks, likely shattered by freeze–thaw processes, are scattered on the floor (Moseley et al., 2020). Accumulations of CCMs lie on the shattered flowstone blocks in an area ca. 3 x 1 m. At this location, an air temperature of -14.7°C was measured in July 2019 (Fig. 2), while the outside air temperatures reached up to 18°C (Donner et al., 2020; Moseley et al., 2020). Measurements of relative
humidity only exist for other parts of the cave where temperatures were above -10°C, reaching values as low as 39 % at floor level (Barton et al., 2020). It can therefore be inferred that the relative humidity in colder parts of the cave must also be rather low. Cold air has less capacity to hold moisture and incoming air loses much of its humidity by resublimation, creating hoar frost on the walls close to the cave entrance (Barton et al., 2020; Lauriol et al., 1988). Deeper inside the cave, flowstone deposits block a ca. 5 m deep vadose slot, at the bottom of which cave air temperatures reached -17.1°C (Barton et al., 2020; Moseley et al., 2020). The
cause of these low temperatures is likely density-driven flow of very cold winter air into this descending single-entrance cave. During summer, a stable air stratification inside this cold trap prevents advection of warm outside air into the cave (Barton et al., 2020).



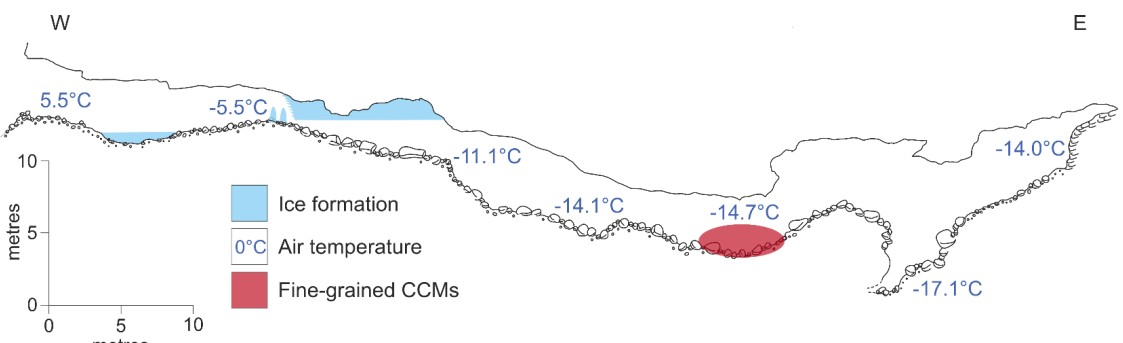

**Figure 2**: Longitudinal section of Cove Cave. Ice formations such as an ice pond, ice stalagmites, and hoar frost are only present in the first third
of the cave. Beyond, air temperatures are much lower. CCMs were found ca. 65 m behind the entrance (adapted from Moseley et al., 2020).

### 3 Materials and Methods

#### 3.1 Sampling and sample preparation

Sampling took place in four different spots within the 3 x 1 m accumulation of CCMs in July 2019. Four samples (KC19CCC-1,
KC19CCC-2, KC19CCC-3, and KC19CCC-4) were picked up with a knife, wrapped in aluminium foil, and stored in plastic
containers. During sampling, disturbance to the overall appearance of the CCM accumulation and Cove Cave was kept to a
minimum.

In the lab, the samples were homogenised and transferred to glass vials in a laminar-flow hood. For all subsequent analyses,
macroscopic contaminants (e.g., pieces of bedrock or insect remains) were removed under a binocular microscope. To aid the
calculation of the DCF, four artificial mixtures ranging from 100 wt.-% CCM fraction to 100 wt.-% non-CCM fraction were
produced from KC19CCC-4. Overall, analysis options and the reproducibility of analyses were limited by the available sample
amount.

#### 3.2 Mineralogy and crystal morphology

A Bruker D8 Discover X-ray diffractometer (XRD) in Bragg-Brentano geometry equipped with a Cu target and a LYNXEYE
detector was used to analyse the mineralogical composition of the samples. Further, a HORIBA JOBIN-YVON LabRam-HR800
spectrometer, excited by a frequency-doubled Nd-YAG laser (100 mW, 532 nm), was used for the in situ determination of the
mineralogy of individual grains at a resolution of ca. 5 µm.

A Keyence VHX 6000 digital microscope was used to examine the morphology of the CCMs. To study the fine fraction, a field-
emission scanning electron microscope (SEM) operating at 10 kV accelerating voltage was utilised (DSM 982 Gemini, Zeiss).

#### 3.3 Stable isotope analysis

Fourteen aliquots of 0.15 to 0.7 mg of the carbonate fraction were taken from the four samples and their carbon and oxygen stable
isotopic composition was analysed using a Thermo Fisher Delta V Plus isotope ratio mass spectrometer coupled with a Gasbench



II (Spötl and Vennemann, 2003), yielding a long-term precision of ±0.08 ‰ for δ[18]O and ±0.06 ‰ (1 σ) for δ[13]C (Spötl, 2011).
For comparison, carbon and oxygen isotope data from inactive common speleothems in the study area were included. All results
were calibrated against international standards and reported relative to the Vienna Pee Dee Belemnite (VPDB) standard.

**3.4 Radioisotope dating**

**3.4.1 [230]Th/U disequilibrium dating and isochron construction**

For [230]Th/U disequilibrium dating of the carbonate fraction, 20 mg aliquots from each of the four samples were picked in a laminar-
flow hood. Dating, including chemical preparation and multi-collector inductively coupled mass spectrometry, was carried out at
the Trace Metal Isotope Geochemistry Laboratory at the University of Minnesota following Edwards et al. (1987) and Shen et al.
(2012). Ages are reported in years before 1950 CE (a BP) with 2 σ uncertainties. Additionally, δ[234]U values of inactive common

speleothems from Cove Cave were included in this study.

An isochron using maximum likelihood regression (Ludwig and Titterington, 1994) was constructed in IsoplotR (Vermeesch,
2018) using the [238]U-normalised activities of [234]U, [230]Th and [232]Th. Additionally, the initial [230]Th/[232]Th activity ratio was ascertained
and applied to the detrital Th correction of the individual ages as a reliability test of the derived isochron values in order to establish
whether there was one or multiple sources of detrital [230]Th (Dorale et al., 2004). Ages are reported in years BP with 2 σ

measurement uncertainties.

**3.4.2 Radiocarbon dating and calculation of DCF**

Eight aliquots of 16–31 mg were taken from the samples, including artificial mixtures. All eight aliquots were analysed with a
MICADAS accelerator mass spectrometer (AMS) in the Laboratory of Ion Beam Physics at ETH Zurich (Synal et al., 2007).

Radiocarbon values are reported as conventional radiocarbon ages before present ±1 σ (BP) (Stuiver and Polach, 1977) and as
fraction modern ±1 σ (F[14]C) (Reimer et al., 2004; Reimer et al., 2020) relative to 95 % of the secondary standard HOx2 (Hajdas et
al., 2021).

The DCF calculation is based on the equation given by Genty and Massault (1997), using the measured F[14]C of the sample
(F[14]C$_{sample}$) in relation to the atmospheric F[14]C at the time of formation (F[14]C$_{atm}$), which was obtained by the corrected [230]Th/U age

and the IntCal20 calibration curve (Reimer et al., 2020; Eq. 1):

$$DCF = \left(1 - \frac{F^{14}C_{sample}}{F^{14}C_{atm}}\right) \times 100\ \%$$
(1)

The radiocarbon ages were then corrected for the corresponding DCFs, resulting in a radiocarbon age ±2 standard errors for 0 %
DCF, calculated by linear approximation and calibrated against IntCal20 (cal BP) (Reimer et al., 2020) using OxCal 4.4 (Bronk
Ramsey, 2009).


**4 Results**

**4.1 Mineralogical composition and particle morphology**



All four samples show a complex mineralogy and a range of particle morphologies that consist of crystal aggregates and single crystals of variable size. The aggregates range from 50 to 400 µm in diameter, rarely exceeding 500 µm, while individual crystals
range from 1 µm to a few tens of micrometres. Brownish sub-micrometric crystals dominate sample KC19CCC-1, which contains larger translucent and opaque white as well as brownish crystals and crystal aggregates (Fig. 3a). Sample KC19CCC-2 (Fig. 3b) is homogeneously white with opaque and translucent crystals/crystal aggregates. The other two samples (KC19CCC-3 and -4; Fig. 3c) are made up of larger white and brownish crystals/crystal aggregates and brownish sub-micrometric crystals resulting in a speckled appearance.

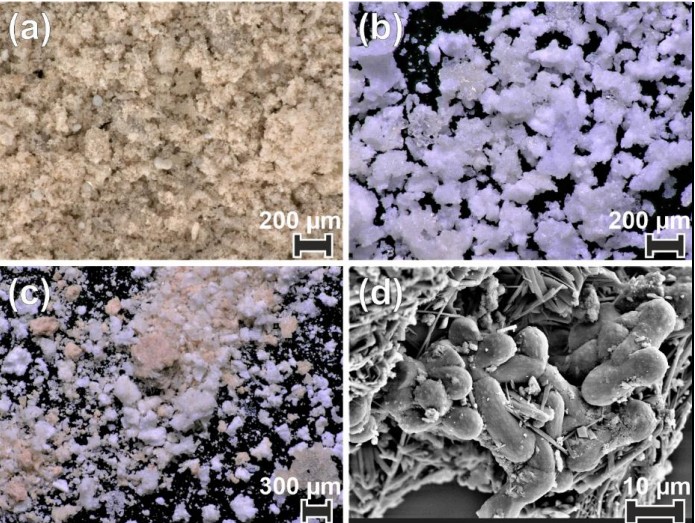

**Figure 3**: Optical (a–c) and SEM (d) images of the studied cryogenic samples. (a) Brownish very fine crystals (unknown composition) intermixed with brownish crystal aggregates (calcite) and white/translucent single crystals made up of calcite, dolomite and gypsum (sample KC19CCC-1). (b) White calcite, gypsum, eugsterite, mirabilite, and löweite (sample KC19CCC-2). (c) Mixed white (calcite and sulphate minerals) and brownish (calcite) particles (samples KC19CCC-3 and -4). (d) Intermixed dumbbell-shaped calcite and acicular sulphate minerals which are present in all
samples.

XRD results show that the samples are mostly mixtures of calcite and the sulphate minerals gypsum ($CaSO_4 \cdot 2H_2O$), eugsterite ($Na_4Ca(SO_4)_3 \cdot 2H_2O$), löweite ($Na_{12}Mg_7(SO_4)_{13} \cdot 15H_2O$), and mirabilite ($Na_2SO_4 \cdot 10H_2O$). Sample KC19CCC-1 contains additional phases such as quartz, dolomite, and potassium feldspar.

The CCMs contain a variety of crystal morphologies, comparable to the morphologies of $CCC_{coarse}$ (Žák et al., 2018), albeit of
much smaller sizes. Predominant morphologies are spherulitic and acicular as well as aggregates of elongated prismatic crystals. Further, rhombic and dumbbell-shaped morphologies were observed. Optical microscopy combined with micro-Raman spectroscopy showed that calcite is mostly represented as translucent and opaque spherulitic crystals and crystal aggregates such as chains of different colours, whereas sulphate minerals mostly consist of white fibrous, acicular, and prismatic crystals. Raman spectroscopy and SEM imagery also showed that microcrystalline calcite and sulphate minerals are often intermixed (Fig. 3d).


### 4.2 Stable isotopes

The stable isotopic composition of the 14 aliquots ranges from -24.0 to -16.0 ‰ for $\delta^{18}O$ and from 7.0 to 11.4 ‰ for $\delta^{13}C$ (Fig. 4). The $\delta^{13}C$ values overlap with those of $CCC_{fine}$ from mid-latitude caves (Žák et al., 2018). The $\delta^{18}O$ values, however, are





significantly lower than values from mid-latitude CCC$_{fine}$ and fall at the lower end of δ$^{18}$O values of mid-latitude CCC$_{coarse}$ (Žák et

al., 2018).

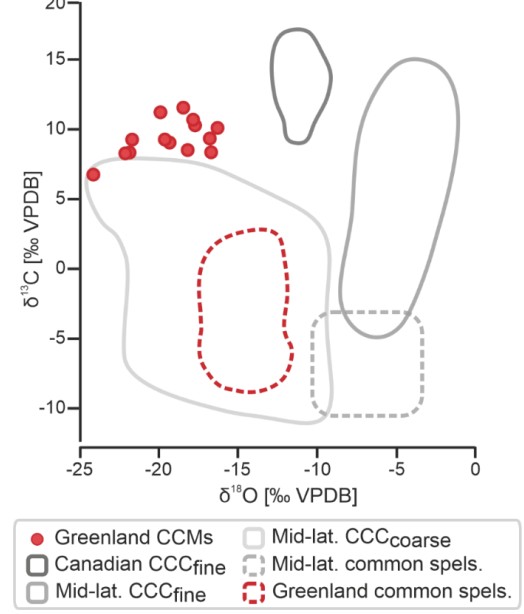

**Figure 4**: Stable isotope composition of CCMs and common speleothems from northeast Greenland (this study) compared to CCC$_{fine}$ from Canada (Clark and Lauriol, 1992) as well as CCC$_{fine}$ and common speleothems from mid-latitude caves (Žák et al., 2018). The composition of CCC$_{coarse}$ from mid-latitude caves is also shown for comparison (Žák et al., 2018). Note the difference in δ$^{18}$O between mid- and high-latitude

CCC$_{fine}$/CCMs as well as between mid-latitude and Greenland common speleothems.

### 4.3 Radioisotope dating

Of the four $^{230}$Th/U analyses, three yielded results (KC19CCC-2, KC19CCC-3, KC19CCC-4), while KC19CCC-1 contained too much detrital material to be analysed. Table 1 shows that the $^{238}$U content is high (1113±2 to 1278±2 ng g$^{-1}$), as is the concentration

of detrital thorium (indicated by the low $^{230}$Th/$^{232}$Th activity ratio between 1.30±0.04 and 3.89±0.22). Uncorrected ages show a large range from 143±11 to 7335±48 a BP. However, the appearance of the sampling site and the presence of the same types of minerals in all samples suggest that they likely formed near-synchronously. The results yielded a highly correlated isochron (R²=0.998), yet the maximum likelihood regression is heavily controlled by the sample with the highest analytical precision, which is the cleanest sample (KC19CCC-2). The high mean square of the weighted deviates (MSWD) of 69 indicates that the ages are

over-dispersed compared to the stated analytical uncertainties and that an isochron accounting for this overdispersion is needed (Vermeesch, 2018). The resulting age of 65±17 a BP agrees with the ages of two of the three samples (KC19CCC-2: 64±12 a BP; KC19CCC-3: 172±153 a BP) when corrected for detrital Th using the isochron-derived initial $^{230}$Th/$^{232}$Th activity of 1.39±0.006 (Fig. 5). The isochron-corrected age of KC19CCC-4 (353±74 a BP) is, however, not in agreement with the other two samples within dating uncertainty, thus suggesting multiple sources of initial $^{230}$Th (e.g., Dorale et al., 2004). Nonetheless, this disagreement

could also result from an underestimation of uncertainty by the maximum likelihood regression. The isochron-corrected δ$^{234}$U$_{initial}$ is high for all three samples (1872±3 to 2193±5).





**Table 1**: Results of $^{230}$Th/U disequilibrium dating.

| Sample | $^{238}$U (ng g$^{-1}$) | $^{232}$Th (ng g$^{-1}$) | $^{230}$Th/$^{232}$Th (activity) | δ$^{234}$U* (measured) | $^{230}$Th/$^{238}$U (activity) | Uncorrected $^{230}$Th age (a) | δ$^{234}$U$_{initial}$ | Bulk-earth-corrected age (a BP) | Isochron-corrected age (a BP)† |
|---|---|---|---|---|---|---|---|---|---|
| KC19CCC-1 | 2658 ±6.5 | 2028±40.9 | 2.04±0.04 | 9±1.9 | 0.5139 ±0.0033 | | | | |
| KC19CCC-2 | 1236 ±3.0 | 6±0.1 | 3.89±0.22 | 2192±5.1 | 0.0062 ±0.0003 | 213±11 | 2193±5 | 97±34 | 64±12 |
| KC19CCC-3 | 1113 ±1.9 | 480±9.6 | 1.30±0.04 | 1871±3.4 | 0.1898 ±0.0012 | 7405±48 | 1872±3 | 2918 ±3140 | 172±153 |
| KC19CCC-4 | 1278 ±1.7 | 270±5.4 | 1.48±0.04 | 2058±2.9 | 0.1049 ±0.0007 | 3792±25 | 2058±3 | 1700 ±1433 | 353±74 |

All uncertainties are 2 σ.
U decay constants: λ$_{238}$ = 1.55125x10$^{-10}$ (Jaffey et al., 1971) and λ$_{234}$ = 2.82206x10$^{-6}$ (Cheng et al., 2013). Th decay constant: λ$_{230}$ = 9.1705x10$^{-6}$ (Cheng et al., 2013).
*δ$^{234}$U = (($^{234}$U/$^{238}$U)activity − 1)x1000.
†corrected for detrital Th with initial $^{230}$Th/$^{232}$Th activity of 1.39±0.006

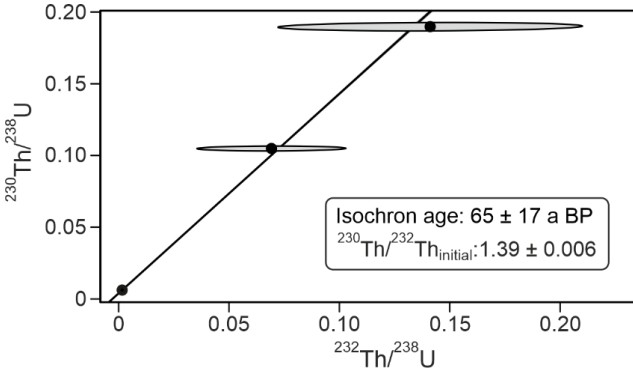


**Figure 5**: Maximum likelihood isochron with overdispersion based on the three $^{230}$Th/U disequilibrium ages and calculated in IsoplotR (Vermeesch, 2018).

Conventional radiocarbon ages for the four untreated and four mixed aliquots range from 140±27 to 2261±74 a BP, with corresponding F$^{14}$C values of 0.98±0.003 to 0.06±0.0006 (Table 1). When applying the radiocarbon method to speleothems the

DCF needs to be determined (Genty and Massault, 1997) in order to correct for $^{14}$C-free material from bedrock and/or soil (Bajo et al., 2017; Hajdas et al., 2021; Hua et al., 2012). The DCFs of all analysed aliquots range from 1.24±0.50 to 93.67±0.06 % (1 σ uncertainty). For the untreated aliquots, 96.2 % of variance in radiocarbon age can be explained by the DCF (R²=0.96). When artificial mixtures are included, this value is lower (R²=0.88), yet indicating that the artificial mixing was reasonably accurate. The radiocarbon age extrapolated to 0 % dead carbon by linear approximation is 2±28 BP. Calibration using IntCal20 (Reimer et al.,

2020) yielded three distinct peaks of calibrated ages: 36–73, 112–139 and 226–255 cal BP with respective probabilities of 37.8, 28.3 and 29.4 % (Fig. 6). The peak with the highest probability (36–73 BP) is in agreement with the $^{230}$Th/U isochron age of 65±17 a BP and provides the most likely timing for CCM formation. We attribute the other two peaks to a plateau of the calibration curve in this time interval.

**Table 2**: Results of $^{14}$C dating.

| Sample | Lab code | $^{14}$C age (BP) | | F$^{14}$C | | δ$^{13}$C (‰) | | DCF (%) | |
|---|---|---|---|---|---|---|---|---|---|
| KC19CCC-1 | ETH-109243 | 12562 | ±33 | 0.209 | ±0.0009 | 6.7 | ±1 | 79.00 | ±0.12 |
| KC19CCC-2 | ETH-109244 | 140 | ±27 | 0.983 | ±0.0033 | 9.0 | ±1 | 1.24 | ±0.50 |
| KC19CCC-3 | ETH-109245 | 6337 | ±35 | 0.454 | ±0.0020 | 3.1 | ±1 | 54.39 | ±0.26 |
| KC19CCC-4 | ETH-109246 | 1626 | ±28 | 0.817 | ±0.0028 | 7.9 | ±1 | 17.92 | ±0.42 |





| | | | | | | | | | | |
|---|---|---|---|---|---|---|---|---|---|---|
| **KC19CCC-4**<br>100 % clean | ETH-112579 | 358 | ±22 | 0.956 | ±0.0026 | 10.6 | ±1 | 3.95 | ±0.45 |
| **KC19CCC-4**<br>90 % clean | ETH-112580 | 1758 | ±23 | 0.803 | ±0.0023 | 9.5 | ±1 | 19.33 | ±0.38 |
| **KC19CCC-4**<br>70 % clean | ETH-112581 | 2353 | ±23 | 0.746 | ±0.0021 | 8.8 | ±1 | 25.05 | ±0.35 |
| **KC19CCC-4**<br>0 % clean | ETH-112582 | 22261 | ±74 | 0.063 | ±0.0006 | 2.2 | ±1 | 93.67 | ±0.06 |
| All uncertainties are 1 σ. | | | | | | | | | |


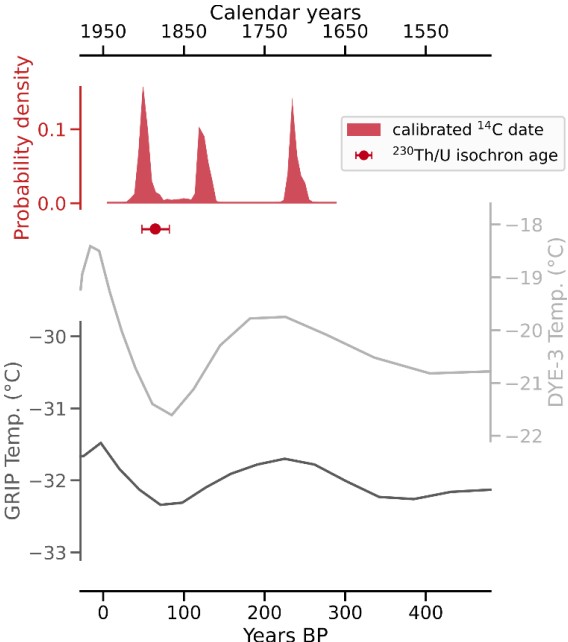

**Figure 6**: Calibrated and DCF-corrected [14]C dating results and [230]Th/U isochron age plotted against temperatures from two boreholes on the Greenland ice sheet (Dahl-Jensen et al., 1998) indicate that the Cove Cave CCMs formed during the late Little Ice Age. The two older peaks of the calibrated and corrected [14]C dates are likely the result of a plateau of the calibration curve at that time interval.

The dating results show that the CCMs formed during the late Little Ice Age, a period of relatively cold climate conditions and glacier advances, which in Greenland lasted between about 700 and 50 a BP (i.e., 1250 – 1900 CE; Kjær et al., 2022).

## 5 Discussion

### 5.1 Formation of CCMs

In carbonate-hosted caves, $CCC_{fine}$ typically show a simple mineralogical composition, mostly consisting of calcite or, less commonly, other carbonate minerals such as aragonite, or mixtures thereof (Žák et al., 2018). In some caves, the occurrence of non-carbonate minerals of cryogenic origin has been reported (Dublyansky et al., 2017; Žák et al., 2018). Cryogenic gypsum and, locally, other sulphate minerals are mostly known from caves hosted in gypsum rock (Kadebskaya and Tschaikovskiy, 2015; Žák et al., 2018) but also from some limestone caves where the sulphate is derived from pyrite oxidation (Bartolomé et al., 2022;

Dublyansky et al., 2017). Mirabilite is known from several caves where its existence has been tied to evaporative conditions (e.g., Audra and Nobécourt, 2013; Bieniok et al., 2011). Although it has been reported from cold caves (Harmon et al., 1983) and its formation has been tied to preceding cryochemical processes, mirabilite was not regarded as primarily cryogenic in origin (Žák et





al., 2018), and we are unaware of reports about cryogenically formed mirabilite in caves. Eugsterite is a rare sulphate mineral which has only been reported from two caves, Mammoth Cave, USA (White, 2017), and Chamois Cave, France (Audra and

Nobécourt, 2013), where its formation is not related to cryogenic processes. Löweite has only rarely been described from volcanic caves (Hill and Forti, 1997).

In Cove Cave, the most likely source of dissolved sulphate, necessary for the precipitation of sulphate minerals, is finely disseminated pyrite in the dark-coloured Silurian limestones in which this cave developed (Smith and Rasmussen, 2020), whereas magnesium for löweite formation was likely sourced from dolostones in the area. In contrast the presence of sodium, which is

needed for the minerals mirabilite, löweite, and eugsterite is more challenging to explain. One possibility is that, given the proximity to the coast, the Na is sourced from aerosols from the ocean and/or the sea ice surface, which has been recognised as a dominant source for Na in coastal Arctic ice cores (Rhodes et al., 2018).

The mode of formation (during freezing or sublimation) of the hydrated sulphate minerals must also be considered, especially with respect to whether their formation was synchronous with the precipitation of $CCC_{fine}$. The high solubility of eugsterite, mirabilite,

löweite, and to a lesser extent gypsum, can however be used to provide such insights. For instance, it is clear that the four sulphate minerals did not form prior to the influx of water and subsequent freezing, otherwise pre-existing soluble minerals would have been dissolved and/or washed away. Likewise, the high solubility of these sulphate minerals suggests that the ice in which the $CCC_{fine}$ were embedded, did not melt but rather sublime, otherwise these delicate crystals would not have been preserved. Furthermore, the cave section hosting the sampling site must have remained dry and ice-free since the formation of the CCMs.

This also implies that the cave ventilation pattern at the time of formation was already similar to the present mode in this sag-type cave.

Based on the observation that the calcite and sulphate minerals were found intermixed, it seems likely that the formation of the sulphate minerals was associated with the formation of $CCC_{fine}$, in the sense that both their formations were triggered by the same event, and therefore took place near-synchronously. Whether this formation was cryogenic, or occurred during the subsequent

sublimation of ice cannot be determined with certainty. A cryogenic formation of gypsum is likely, similar to the observed formation in other ice caves (see above; e.g., Žák et al., 2018) but the other sulphate minerals may have formed during the subsequent sublimation of ice after gypsum and $CCC_{fine}$ had already formed.

### 5.2 Stable isotopes

The relatively high $\delta^{13}C$ (7.0-11.4 ‰) values of the Greenland CCMs are comparable to those of $CCC_{fine}$ from the Canadian Arctic circle (Clark and Lauriol, 1992), however, the $\delta^{13}C$ of the Canadian samples reach higher values (up to 17 ‰). Nevertheless, these highly positive values reflect kinetic isotope fractionation as a result of rapid freezing and associated degassing of carbon dioxide (Lacelle et al., 2009).

The Cove Cave CCMs show highly depleted $\delta^{18}O$ (-24.0 to -16.0 ‰) values that overlap with the lower range of $\delta^{18}O$ values of

$CCC_{coarse}$ from mid-latitude caves (Fig. 4). Whilst $CCC_{coarse}$ $\delta^{18}O$ reflects closed-system freezing in small drip water pools in ice, the low values of the Greenland samples are largely related to the isotopically light meteoric precipitation in this high-Arctic setting. This shift is also seen in $CCC_{fine}$ from Canadian caves (Clark and Lauriol, 1992; Fig. 4), although at a smaller magnitude. There is also a difference between the $\delta^{18}O$ values of CCMs and (inactive) common speleothems from the same cave (-24 to -16 ‰ vs. -17 to -13 ‰ respectively), which is not in agreement with other $CCC_{fine}$ vs. common speleothem studies, where the $\delta^{18}O$



values of CCC$_{fine}$ either overlap with those of common speleothems or are slightly higher (Fig. 4; e.g., Luetscher et al., 2013; Žák et al., 2018).

### 5.3 Formation age

In this study, CCC$_{fine}$ were successfully dated using both a U-Th isochron approach and $^{14}$C dating with DCF correction. The
isochron indicated that the the initial $^{230}$Th/$^{232}$Th activity ratio (1.39±0.006) was elevated in comparison to the bulk earth-derived value (i.e., 0.8; Wedepohl, 1995), which had therefore resulted in an under-correction for detrital Th (Table 1). For dating CCC$_{fine}$ in this study, it is sufficient to solely use $^{230}$Th/U methods but we also show that successful and reliable dating of CCC$_{fine}$ is possible using $^{14}$C provided that the DCF can be constrained and corrected for. The independent age information, necessary for the DCF calculation, however, does not have to be provided by $^{230}$Th/U dating; it could for instance come from $^{14}$C dating of stratigraphically
coeval organic matter. Where $^{230}$Th/U dating of CCC$_{fine}$ is impracticable (e.g., due to low $^{238}$U concentrations and/or high detrital Th contamination), $^{14}$C dating could provide a viable alternative. In this study, the $^{14}$C dating results complement the $^{230}$Th/U isochron age well. However, because the DCF-corrected $^{14}$C age hits a plateau on the calibration curve (Reimer et al., 2020), the $^{230}$Th/U method is considered a better approach in this study. On the other hand, $^{14}$C dating of CCC$_{fine}$ could be equally reliable as, or better than $^{230}$Th/U dating for periods of time characterised by a steeply sloped calibration curve. Ultimately, the choice of dating
will be study-dependent and regardless of the method used, dating of CCC$_{fine}$ could provide additional chronological control on cave ice bodies, an emerging (but rapidly disappearing) palaeoclimate archive (e.g., Kern and Perşoiu, 2013; Racine et al., 2022), in particular in those settings were organic inclusions are lacking.

### 5.4 Extreme event triggered CCM formation

High δ$^{234}$U$_{initial}$ values suggest that the karst system of the cave was hydrologically inactive prior to the event that caused CCM formation. During arid climate periods and/or under permafrost influence, $^{234}$U accumulates in the crystal lattice of bedrock minerals through alpha recoil and is easily mobilised when water becomes available (Fleischer, 1982). Speleothems, including CCMs, may record this first mobilisation of $^{234}$U after a period of accumulation by elevated δ$^{234}$U$_{initial}$ values (e.g., Wendt et al., 2020). The high δ$^{234}$U$_{initial}$ of the Cove Cave CCMs (1872±3 to 2193±5) contrasts with the comparatively low δ$^{234}$U$_{initial}$ of common
speleothems from the same cave, which represent a hydrologically active karst system (<150). The high δ$^{234}$U$_{initial}$ of the Cove Cave CCM therefore indicate a prolonged period of permafrost presence prior to their formation.

Combining the results of mineralogical analyses and dating, it can be concluded that the Cove Cave CCMs formed as a result of a singular event at 65±17 a BP (1889±17 CE) and therefore during the cold climate of the late Little Ice Age (ca. 700-50 a BP; Kjær et al., 2022). A reconstruction of surface temperature based on Greenland ice core data since 1840 CE indicates that the mid- to
late-1880s CE (65–60 a BP) were some of the coolest years on record (Box et al., 2009). It can therefore be assumed that the climate of northeast Greenland during the time of CCM formation was colder and permafrost was more abundant than today. In order for the water to enter the cave through the frozen rock and not immediately freeze and sublime on the walls, it must carry enough latent heat. Any event that led to these conditions in and around Cove Cave must have been anomalous, which is supported by the lack of CCMs observed in other caves of the area.

In order to explain the formation of fine-grained CCMs in Cove Cave, we consider the following scenarios: i) an extreme rainfall event providing enough water and thus latent heat to enter Cove Cave; ii) migration of the hoar frost boundary and melt pond





deeper into the cave due to the influx of warm and moist air; iii) an increase of the active layer thickness of (wet) permafrost due to an anomalously warm summer/year/interval; iv) higher cave air temperatures and water availability due to temperate ice covering the cave; and v) enhanced melting of the local ice cap.

There is no evidence for (i) an extreme rainfall event in the high-resolution Greenland ice cores during the time of CCM formation, although it cannot be excluded that such an event occurred locally or regionally. Though a tectonic fracture exists in the roof of Cove Cave, which could promote water infiltration, the lack of CCM formation in other caves of the area renders this scenario unlikely. Furthermore, in this arid region with ca. 200 mm per year of precipitation (Schuster et al., 2021), extreme rainfall events with high volumes of water do not occur. Scenario (ii) seems unlikely as the hoar frost boundary is located at the high point in

Cove Cave (Fig. 2). Such boundaries are only observed at the highest points in the Greenland Caves (Barton et al., 2020), hence shifting this boundary deeper into the cave is unlikely to occur based on the sag-type geometry of the cave, which determines the location of the cold pool interface. An unusually warm summer/year/interval that could potentially lead to (iii) an increase of active layer thickness would have been recorded by Greenland ice cores as well as observational records along the coast of south and west Greenland. Based on a merged south-west Greenland temperature record (Vinther et al., 2006), there is no

summer/year/interval that stands out as being particularly warm in the period of 1868-1902 CE (i.e., 65±17 a BP). Another argument against scenario (iii) is, again, the lack of CCMs in other caves in the area. While it is very likely that an ice cap covered the plateau above Cove Cave in the time interval of interest (Fig. 1), scenario (iv) fails to explain why there was only one generation of CCMs found in Cove Cave, since higher cave air temperatures and water availability would probably have lasted for a prolonged interval. The ice cap does, however, appear to be a key factor together with a tectonic fracture in the roof of the cave, as they enable

a mechanism in which sufficient water can pass through the permafrost and enter the underground. Specifically, enhanced melting of the local ice cap (v) could have been triggered by anomalously high temperatures that lasted a few days (Neff et al., 2014) and concurrent lowering of the albedo due to black carbon deposition on the ice cap. This combination of factors led to ice-melting conditions all over the Greenland ice sheet in 1889 CE (Clausen et al., 1988; Fischer et al., 1998; Keegan et al., 2014; Neff et al., 2014), which is often referred to as the "summer melt episode" of 1889 CE. This episode occurred synchronously within dating

uncertainty with the timing of the CCM formation (1885 CE; 65±17 a BP). Closer to our study site, an ice core from the Flade Isblink ice cap recorded a warming between 1920–1930 CE (30–20 a BP) as increased melt percentage, but it did not record enhanced melt during 1889 CE, which might simply be the result of a non-definitive time scale that does not allow for the investigation of short-term events (Lemark, 2010). Nevertheless, we infer that the unusually high air temperatures and black carbon deposition associated with this event affected our study area as well, leading to enhanced melting of the local ice cap, with more

water entering moulins, reaching the base of the ice cap, and then finding its way through conduits, possibly through tectonic fractures, into the cave. This water turned to ice in the heavily undercooled cave resulting in the precipitation of $CCC_{fine}$ and potentially also cryogenic gypsum during a rather rapid freezing process. Subsequently, the ice sublimated in the cold and dry microclimate of the cave releasing enclosed $CCC_{fine}$ and cryogenic gypsum particles which accumulated on the cave floor. As discussed above, the origin of the other hydrous sulphate minerals is less clear; they were, however, found intermixed with $CCC_{fine}$

and gypsum in the same spot, hinting towards a near-synchronous formation.

**6 Conclusions**

Fine-grained CCMs from Cove Cave, a low-elevation, high-latitude permafrost cave in northeast Greenland, consist of an unusual mixture of minerals: cryogenic calcite ($CCC_{fine}$), potentially cryogenic gypsum, and other hydrous sulphate minerals (eugsterite,

mirabilite, löweite), whose mode of formation is less clear but presumably at least associated with cryochemical processes and/or subsequent sublimation of the ice body in a dry cave atmosphere. The high solubility of the sulphate minerals suggests that since their formation, and after the sublimation of ice, the microclimate at the sampling site has remained cold, dry, and ice-free, as it is today.

The Greenland CCMs extend the knowledge of the stable isotopic composition of CCMs. While their $\delta^{13}$C values are comparable
to those of mid-latitude CCC$_{fine}$, the $\delta^{18}$O values are lower compared to both common (inactive) speleothems from the same cave and CCC$_{fine}$ from the mid-latitudes. These lower $\delta^{18}$O values can be attributed to much lower $\delta^{18}$O values of meteoric precipitation in the high latitudes and, to a lesser extent, the difference in the isotopic composition of the water sources during the formation of CCMs and common speleothems.

We show that precise dating of CCC$_{fine}$ is in principle possible with $^{230}$Th/U or $^{14}$C dating. Using isochrons and a site-specific
initial $^{230}$Th/$^{232}$Th correction factor is paramount for accurate $^{230}$Th/U ages. In this study, $^{230}$Th/U is arguably superior to $^{14}$C dating, which is dependent on the slope of a calibration curve and independent age information to constrain the DCF. However, its application is restricted to samples of low detrital Th content (and rather high U concentration). Where these conditions cannot be met, $^{14}$C dating might be a better alternative. The possibility to date CCC$_{fine}$ with either $^{230}$Th/U or $^{14}$C could aid the establishment of reliable chronologies for cave ice as a palaeoclimate archive.

We conclude that CCM formation in Cove Cave was most likely the result of a single extreme event leading to the summer melt episode of 1889 CE on the Greenland ice sheet, which also caused enhanced melting on the local ice cap on the plateau above the cave.

**Competing interests**

The authors declare that they have no conflict of interest.

**Author contribution**

AD designed the methodology, interpreted the data, and wrote the paper. PT assisted with field work, age calculations, and manuscript preparation. CS assisted with analyses, contributed to discussions and manuscript preparation. IH performed $^{14}$C dating.
XL contributed with $^{230}$Th/U disequilibrium analyses. RLE provided analytical $^{230}$Th/U disequilibrium dating facilities. GEM (P. I.) designed the study, raised the funding, organised, led and participated in the fieldwork, contributed to the discussions and manuscript preparation.

**Acknowledgements**

We thank Manuela Wimmer for stable isotope measurements, Clivia Hejny for XRD analyses, Kristian Pfaller for SEM images, Andreas Saxer for XRD analyses and SEM images, and Bastian Joachim-Mrosko for Raman spectroscopy. We are grateful to Hazel Barton, Chris Blakeley, Pete Hodkinson, Adam Ignézi, Robbie Shone, Paul Smith, and Andrew Sole, who participated in the 2019 Greenland Caves Expedition. This research has been supported by the Austrian Science Fund (FWF) project no. Y 1162N37 to Gina E. Moseley. Data from the Programme for Monitoring of the Greenland Ice Sheet (PROMICE) were provided



by the Geological Survey of Denmark and Greenland (GEUS) at http://www.promice.dk. The Greenland government are thanked

for permission to undertake this fieldwork (KNNO Expedition Permit C-19-32; Scientific Survey Licence VU-00150; Greenland

National Museum and Archives 2019/01).

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
