# Peer review of "Cryogenic cave minerals recorded 1889 CE melt event in northeast Greenland"

_Climate of the Past, 2022_

## Author Comment (AC1)

**Comments by reviewer #1**

*We thank reviewer #1 for their constructive comments, which will improve the manuscript. Our response to the individual comments including proposed changes to the manuscript are given below.*

In this paper, the authors present evidence of fine-grained cryogenic cave minerals (CCMs) found in Cove Cave in northeast Greenland. They provide evidence for their ability to date these fine-grained CCMs, unlike previous studies, and ultimately demonstrate that these CCMs likely formed during a short, but extreme, melting event on the Greenland Ice Sheet in 1889, which would have provided sufficient liquid water in the cave to form these minerals.

I'm not an expert in geochemistry, or dating, so my review is general and focuses mainly on the plausibility of the 1889 melting event on the Greenland Ice Sheet providing sufficient meltwater to the Cove Cave system.

General Comments:

1. The authors describe how CCMs are a paleo-permafrost proxy and their importance to mid- and lower-latitude sites. What do the presence of CCMs in this northern-most location tell us about permafrost in this high-latitude site? It's clearly interesting that these CCMs appear to have formed during a short-term extreme event on the ice sheet in 1889. A little more context of why this is an important finding in the Discussion and Conclusion sections would be helpful for the broad audience of this journal.

*In contrast to coarse-grained CCMs, the fine-grained form (our samples) cannot be used as a palaeo-permafrost proxy. We will change the phrasing in the introduction so that this point comes across more clearly. However, the presence of CCMs tells us two very interesting things: 1. They show that conditions leading to widespread melting reached the study area in northeast Greenland, 2. The fact that they formed during this short-term extreme event in probably a matter of days, provides hints about the rate of formation (which is a highly understudied field), while also demonstrating that they cannot be used as a permafrost proxy because the presence or absence of permafrost is generally not influenced by short-term weather conditions. We will include these points in the Discussion and Conclusion sections.*

1. The authors present 5 potential scenarios that could have produced the CCMs found at Cove Cave and walk the reader through the logic that eliminates 4 of those scenarios. They then cite publications (Clausen et al., 1988; Fischer et al., 1998; Neff et al., 2014; Keegan et al., 2014), which all describe a widespread melting event that occurred in the dry snow zone of the Greenland Ice Sheet in 1889. They hypothesize that this widespread melting event provided liquid water to the Cove Cave system, which allowed the formation of the CCMs found. Indeed, the studies cited identify a widespread, extreme melting event on the ice sheet that created surface meltwater, but it's initially concerning that the study sites in these references are pretty far from Cove Cave. Looking at available records from closer ice core sites B19 (Hatvani et al., 2022)

and Tunu (Grieman et al., 2018), it doesn't look like it was particularly warm in that region of the ice sheet in 1889 and there wasn't a lot of deposition of vanillic acid (VA; an indicator of forest fire activity) which would suggest a decrease in albedo due to black carbon. The closest ice core record would be from Flade Isblink, and the authors report that this record does not suggest excess melting in 1889, but a recent publication shows that there was a high concentration of black carbon in 1889 at Flade Isblink (Eckhardt et al., 2023). Understandably, the perfect ice core record does not exist to prove the theory that these CCMs were formed due to excess meltwater from the 1889 widespread melting event that occurred at higher elevations on the Greenland Ice Sheet. With the increased concentrations of black carbon at the Flade Isblink site so close to Cove Cave, it does appear that the melting event could have happed in this region too. I suggest including the Flade Isblink black carbon record to bolster this hypothesis.

*Thank you for making us aware of this recent publication. We will certainly include the newly available record from Flade Isblink.*

Specific comments:

Line 21: add 'Cove Cave' before 'CCMs' here to let the reader know you're talking specifically about the Cove Cave CCMs again here

*Will do.*

Line 25: why is 'weather' in parentheses here? I'd suggest removing the parentheses

*We put it in parentheses to also account for the deposition of black carbon but realise that the parentheses might be confusing. We will remove them.*

Lines 53-56: for ease of reading, I suggest breaking this up into two sentences like '…in northeast Greenland. Cove Cave is currently…'

*Will do.*

Line 68: remove 'therefore'

*Will do.*

Line 258: 'sublime' here should be 'sublimated'

*We will change that.*

Line 294: a comma is needed after 'than'

*We will put that in.*

Line 297: 'were' should be 'where' here

*We will change that.*

Figure 1: are the blue lines indicating ice margins referring to present-day ice margins, or ice margins from the former presence of additional ice caps that are mentioned in the last sentence of the figure caption?

*They are indicating ice margins from the former presence of additional ice caps. To make this clearer we will change the figure caption and add "(light blue polygon)" and "(blue lines)" to the respective sentences.*

References:

Eckhardt, S., Pisso, I., Evangeliou, N., Zwaaftink, C. G., Plach, A., McConnell, J. R., ... & Stohl, A. (2023). Revised historical Northern Hemisphere black carbon emissions based on inverse modeling of ice core records. *Nature Communications*, *14*(1), 271.

Grieman, M. M., Aydin, M., McConnell, J. R., & Saltzman, E. S. (2018). Burning-derived vanillic acid in an Arctic ice core from Tunu, northeastern Greenland. *Climate of the Past*, *14*(11), 1625-1637.

Hatvani, I. G., Topál, D., Ruggieri, E., & Kern, Z. (2022). Concurrent Changepoints in Greenland Ice Core δ18O Records and the North Atlantic Oscillation over the Past Millennium. *Atmosphere*, *13*(1), 93.

---

## Author Comment (AC2)

**Comments by reviewer #2 – Connor Turvey**

*We thank Connor Turvey for the helpful and detailed comments. They will certainly improve the manuscript, particularly the section focussing on mineralogy. Please see our response to the individual comments including potential changes to the manuscript below.*

**General Comments**

This paper covers three separate (but interlinked) concepts, it identifies the mineralogy of cryogenic cave minerals (CCM) found in Cove Cave in Greenland, demonstrates that dating information can be extracted from fine grained cryogenic cave carbonates ($CCC_{fine}$, which has proven difficult in other studies) and uses those dating results and other information to infer the circumstances that led to the formation of the CCM in Cove Cave.

Overall this paper seems coherent and well written, I would recommend this paper be accepted pending minor revisions. Detailed comments are provided below.

1. Does the paper address relevant scientific questions within the scope of CP? – Yes it looks at reconstructions of the past by looking at the CCC in a cave in Greenland and also shows how you can get age data from $CCC_{fine}$.

2. Does the paper present novel concepts, ideas, tools, or data? – Yes, age dating of $CCC_{fine}$ is a novel tool.

3. Are substantial conclusions reached? – Yes, it identifies CCM in cove cave and determines their age and formation circumstances.

4. Are the scientific methods and assumptions valid and clearly outlined? - Yes

5. Are the results sufficient to support the interpretations and conclusions? - Yes

6. Is the description of experiments and calculations sufficiently complete and precise to allow their reproduction by fellow scientists (traceability of results)? – Needs improvement. The paper would be improved if they showed more of their mineralogy data (such as XRD diffractograms), and their methods could be more clearly written to allow fellow scientists to use the methods that they outline (eg they need to more explicitly how they are making their mixtures).

*Please see what we plan to change in the specific comments below.*

7. Do the authors give proper credit to related work and clearly indicate their own new/original contribution? – Yes they seem to be citing other relevant work, but they could cite more studies related to the CCM presence/absence of other Greenland caves.

*To the best of our knowledge, there are no other studies on CCMs in Greenland.*

8. Does the title clearly reflect the contents of the paper? - Yes

9. Does the abstract provide a concise and complete summary? - yes

10. Is the overall presentation well structured and clear? -yes

11. Is the language fluent and precise? - yes

12. Are mathematical formulae, symbols, abbreviations, and units correctly defined and used? - yes

13. Should any parts of the paper (text, formulae, figures, tables) be clarified, reduced, combined, or eliminated? - No

14. Are the number and quality of references appropriate? - Yes

15. Is the amount and quality of supplementary material appropriate? – No, we could do with more of the mineralogy either in the text or in the supplementary information.

*We will add the XRD diffractograms to the appendix.*

**Specific Comments**

37 – It would be good to explicitly state the criteria for differentiating $CCC_{fine}$ from $CCC_{coarse}$, presumably a grainsize limit.

*This is a good point. We will add a statement on the differentiating criteria (mostly grain size and isotopic composition as a result of two different formation mechanisms).*

70 – It would be better to report an approximate distance from Cove Cave to the weather stations rather than just saying 'closest'.

*We will add the approximate distances to the weather stations.*

83 – A quick definition for what 'inactive' means in this context might be helpful, presumably it is common in speleothem geology but I am unfamiliar with it.

*We will add a short explanation that could look something like this: "inactive flowstones (i.e., without active water supply)".*

91 – Should probably change "rather low" to something less casual.

*Will do.*

105 – Were there any obvious visual differences (color, texture etc) between the samples during collection?

*No, there were not, all samples were taken from the same accumulation of CCMs and any visual differences were discovered later. We will put that into the text.*

110 – Clarity could be improved here, are you taking sample KC19CCC-4, splitting it into different mineral fractions and then mixing the relative amounts? Or are you mixing KC19CCC-4 with another phase?

*We split KC19CCC-4 into different mineral fractions and then mixed the relative amounts. We will change the phrasing to improve clarity.*

116 – More analytical details for the mineralogy and crystal morphology analysis would be good. For example, with the XRD what was your scan range and analysis time?

*We will add more detail here and add XRD diagrams to the appendix.*

166 – Why could you not ID the very fine brownish crystals? Even if it was too fine to manually separate under a microscope for analysis you had XRD data and could identify the other crystals in the sample so I would have thought it should be possible by process of elimination.

*Based on XRD results, the fine brownish crystals could contain quartz, potassium feldspar or dolomite. We will add that to the text.*

171 – An XRD figure either here or in the appendices showing the results from the 4 samples would be very useful as it would allow for easy comparison between the mineralogy of the four samples, rather than just having it written out.

*As mentioned above, we will add XRD diagrams to the appendix.*

249 – SEM may provide useful insights here, were any textures observed that could only be explained by synchronous formation (crystals intergrown etc)

*We assume that the reviewer is referring to line 254.*

*We carefully examined the SEM images, however, we found no conclusive evidence of coeval crystal growth. We will add that to the text.*

250 – Any idea where does the quartz, dolomite and potassium feldspar in KC19CCC-1 come from? Country rock?

*The most probable source for dolomite in the CCM sample is dolomite, a host rock in the area. We will add this to line 249. Quartz and potassium feldspar are, most likely, detrital material that came into the cave either via water or aeolian transport. We will add this to the manuscript.*

280 – No theory as to why your difference between $\delta^{18}O$ values of your CCC$_{fine}$ vs. common speleothems is different to that seen in other studies?

*Unfortunately, we don't have enough data to draw definite conclusions. Our hypothesis is that the isotopic composition of the source water from which the CCMs and common speleothems precipitated differed. The CCMs were deposited recently but the common speleothems from the same cave (and area) are older and were deposited during an earlier period in the Quaternary with different climatic boundary conditions compared to today. We will not yet disclose the age of the common speleothem samples as those results are intended for another publication and not of relevance here. Previous studies, on the other*

*hand, often compared CCMs and common speleothems of roughly the same geologic age. We will include this in the manuscript.*

314 – Would be good to have a reference supporting your claim that there are not CCMs overserved in other caves in the area.

*To the best of our knowledge, there are no other references that we could cite to support this statement. However, our statement is based on findings from two expeditions that were conducted as part of our project. We will add a sentence regarding this.*

---

## Author Comment (AC3)

**Comments by reviewer #3**

*We thank reviewer #3 for their constructive remarks, which will improve the manuscript. Our response to the individual comments including potential changes to the manuscript are given below.*

This is a very interesting paper that uses unique geologic samples – a cryogenic cave minerals – that grew in a northeast cave of Greenland to reconstruct past climate history. The primary geochemical analyses are mineralogical inspection, stable isotope analyses, and U-Th dating. From these analyses (primarily from the dating results) they deduce that CCMs formed during the Little Ice Age during a period of anomalously high temperatures that occurred over a few days in the summer of 1889 CE. These extreme warm conditions led to widespread melting over the Greenland Ice Sheet.

I am particularly intrigued by the interpretation of "a few days," specifically at LINES 23-24: "We relate the CCM formation to a combination of black carbon deposition and anomalously high temperatures, which occurred over a few days, in the summer of 1889 CE." The time constraint of "days" is an extraordinarily statement – the fact that a few days of extremely high temperatures caused widespread melting over northeast Greenland is an important finding, if it's true. However, I find the author's reasoning for relating CCM growth to this extreme climate event (a few days of warming) insufficient. There is not a thorough explanation for why authors jump to "days"? In the paper, the only citation is Neff et al. (2014). The authors need to add more explanation to this interpretation.

*Please see lines 334-339: The timeframe of the event ("a few days") is not deduced by dating of the Cove Cave CCMs, it stems from several publications that all describe wide-spread melting conditions on the Greenland ice sheet (Clausen et al., 1988; Fischer et al., 1998; Keegan et al., 2014; Neff et al., 2014). The unusual conditions of the summer melt episode of 1889 CE therefore provide the most likely explanation as to why it was possible for CCMs to form during a period of cold climate conditions, while the work of, e.g., Neff et al. (2014) provides constraints ("days"). We will rephrase lines 334-337 to make this point clearer.*

Overall, though, I find this an intriguing paper and I think the authors did a nice job thoroughly explaining their scientific methods and results. I do have a few clarifying points, though, that I think would make the paper stronger. Also, I feel some sections need added details. Most importantly, I find the authors reporting and explanation of the stable isotope data lacking. I explain this more below:

1. *Does the paper present novel concepts, ideas, tools, or data?* – Yes, the dating of CCC material is exceptionally novel, not to mention the location of this cave as the highest-latitude site with paleoclimate data is intriguing

2. *Are the scientific methods and assumptions valid and clearly outlined?* Yes, I think the authors do a nice job clearly stating their scientific processes and methods.

3. *Are the results sufficient to support the interpretations and conclusions?* Yes.

*4. Is the description of experiments and calculations sufficiently complete and precise to allow their reproduction by fellow scientists (traceability of results)?* Yes.

*5. Do the authors give proper credit to related work and clearly indicate their own new/original contribution?* Yes.

*6. Does the title clearly reflect the contents of the paper?* Yes.

*7. Does the abstract provide a concise and complete summary?* Yes, except I recommend removing one part, given it is not relevant to main conclusions.

*8. Is the overall presentation well structured and clear?* Mostly yes, though some sections need more explanation (see my line-by-line comments).

*9. Is the language fluent and precise?* Yes.

*10. Are mathematical formulae, symbols, abbreviations, and units correctly defined and used?* Yes.

*11. Should any parts of the paper (text, formulae, figures, tables) be clarified, reduced, combined, or eliminated?* No.

*12. Are the number and quality of references appropriate?* Yes, except more should be added in reference to the "few days" warm period in 1889 CE.

*Please see our response above.*

*13. Is the amount and quality of supplementary material appropriate?* Yes.

Line-by-line comments

Abstract: I am not sure why authors include the information about $CCC_{fine}$ $\delta^{18}O$ values in the abstract? It is my understanding that they do not use this data to make any interpretations?

*Although we use the stable isotope data to identify which type of CCMs we are dealing with and try to compare our data to those of existing studies, we realise that this information does not have to be included in the abstract. We will remove it.*

Line 39: Though the authors link CCC formation as a "useful proxy for paleo-permafrost," they do not state clearly whether the formation of CCC=permafrost is present? It may be worth stating this explicitly for readers who are unfamiliar with CCC.

*We already state in line 32 that the existence of CCCs indicates past permafrost presence. We will add to the sentence in line 39 that $CCC_{coarse}$ can be used for the reconstruction of past permafrost (i.e., $CCC_{coarse}$ = palaeo-permafrost) as well as negative cave temperatures close to 0°C.*

Line 40: Please state the size difference between $CCC_{coarse}$ and $CCC_{fine}$. Are CCC samples separated into "coarse" and "fine" categories by eye? By measurement?

*They are most commonly differentiated by grain size (~1 mm) but also by their stable isotopic composition, which is the result of the formation mechanism (open vs. closed system). In reality, they seem to occupy a spectrum rather than distinct classes, which is part of an ongoing debate. We will add a statement on the differentiating criteria to the manuscript.*

Line 45: Is there a reason the authors report the CCM subtypes as $CCC_{coarse}$ and $CCC_{fine}$ versus $CCM_{coarse}$ and $CCM_{fine}$? Lines 37-45 explain the difference between CCC and CCM, but then authors refer their CCM samples as CCC? Please clarify, because right now it seems these two are equivalent.

*Unfortunately, the nomenclature in previously published studies is slightly confusing. The term "cryogenic cave minerals" (CCMs) encompasses all kinds of cryogenically formed minerals in caves. These minerals are often carbonates, or more commonly calcite, and therefore abbreviated as CCCs (cryogenic cave carbonates), which themselves can be divided into the subtypes $CCC_{fine}$ and $CCC_{coarse}$ based on their mechanism of formation. We must introduce these terms in the paper to be able to compare our results with existing data from the literature, which mostly focusses on CCCs.*

*Aside from carbonates, CCMs can be composed of other minerals – our samples, of the fine-grained type, are made up of carbonates as well as sulphates. When referring to the carbonate fraction of our samples we use the term $CCC_{fine}$ to differentiate them from the sulphate fraction. We realise, however, that this is confusing to the reader, and we plan to simplify it throughout the paper by dropping the term $CCC_{fine}$ when referring to our samples. We will use the expression "carbonate and sulphate fraction" instead.*

Line 48: "*recently called into question.*" How? Please briefly state this. Perhaps move line 50 up to follow this sentence, since I believe it's because of detrital thorium contamination?

*The accuracy of the $^{230}$Th/U dating technique using standard evaluation procedures has been recently called into question by showing that the conventionally used correction factors for detrital Th contamination are not universally applicable to $CCC_{coarse}$ samples. We will add this to the manuscript. The dating difficulties mentioned in line 50 refer to $CCC_{fine}$.*

In Figure 2, it is interesting you only find fine-grained CCMs in the red shaded region, yet there are other below-0°C regions. Is there a hypothesis for why CCCs formed only in this location in the cave and not in others? This information may be helpful to scientists who want to go and try and find CCMs in other cave systems.

*The conditions under which the Cove Cave CCMs formed are highly unusual (i.e., due to an extreme event under otherwise unfavourable conditions for formation, e.g., too cold and dry). Because the CCMs formed as a result of an extreme event, with contributing factors such as a tectonic fracture in the roof of the cave and an ice cap in close vicinity, the actual cave air temperatures most likely played only a minor or even no role in CCM formation. The location of the CCM accumulation within the cave is probably the result of the location*

*of the fracture, i.e., where water was able to enter the cave, as well as simple settling (and the lack of flowing water) when the ice sublimated and the CCMs accumulated on the floor of the cave.*

Figure 4: It is difficult to discern the difference between the light vs. darker gray shading colors. A suggestion to make one of the categories black?

*Will do.*

Line 125: Please specify where common speleothems were collected. Was it the same cave? Right now it is just reported as "in the study area," which is not enough information. This could help shed light on the $\delta^{18}O$ difference between common speleothem and CCCs?

*The common speleothems were collected in several caves within the study area (a few square km) including Cove Cave. We will add that to the section. This bigger dataset (using isotopic data from the study area) is used to allow for comparison against existing data, i.e., in Fig. 4. In line 278, where we compare $\delta^{18}O$ of common speleothems and of our CCMs, we clearly state that they are both from Cove Cave.*

Section 5.2: Are authors interpreting the low $\delta^{18}O$ values as reflecting contribution of precipitation from the Arctic air mass? Doesn't this location primarily receive precipitation from the Arctic air mass? Why is this significant? Also, the Greenland common speleothems have a higher $\delta^{18}O$ value than CCM $\delta^{18}O$, but they were collected from the same cave? If they are from the same "northeast Greenland" cave, then they should receive precipitation from the same source, and therefore should have the same $\delta^{18}O$? I see at Lines 278-281 the authors address this difference, but do not provide a reason why? Please explain? Even if the authors are not sure why this is, that should be stated. As of now, it is unclear what the assumption of this is, and I find the discussion of the stable isotope data not sufficient.

*We are interpreting the low $\delta^{18}O$ values, which can be observed in both our CCMs as well as the Canadian CCC$_{fine}$, to be the result of the high-latitudinal settings that both sampling areas are located in. We will change this paragraph, focussing more on this latitudinal shift that can be seen in the cryogenic minerals as well as common speleothems, bearing in mind that it would be beyond the scope of this paper to draw major conclusions from this.*

*Regarding the $\delta^{18}O$ values of CCMs and common speleothems from the same cave, the reviewer is correct, they should show the same values if they were of roughly the same age. Our data is not sufficient to draw definite conclusions, however, we hypothesise that the isotopic composition of the source water has changed: while the CCMs were deposited recently, the common speleothems from the same cave were deposited during an earlier period within the Quaternary under different climatic boundary conditions compared to today. We will not yet disclose the age of the common speleothem samples as those results are intended for another publication and not of relevance here.*

*We will change the whole section so that both points come across more clearly. We also plan to add arrows to Fig. 4 to represent the mentioned latitudinal shift.*

Section 5.2 (continued): What is significant about the Greenland CCMs $\delta^{18}O$ overlap with mid-latitude caves? This is not discussed, and I'm a bit confused why this is significant.

*Existing CCC data are strongly biased towards central European/mid-latitude sites. Based on this, the isotopic compositions of $CCC_{fine}$ and $CCC_{coarse}$ are often portrayed to be plotting in distinct O and C isotope ranges. Presenting our data as well as that of Canadian $CCC_{fine}$ indicates that such borders are not universally applicable, especially at higher latitudes. Exploring this further is beyond the scope of this manuscript. We will add a short statement to section 5.2 to improve clarity.*

---

## Author Response (AR1)

Dear Dr Reyes,

We are grateful for the constructive comments and helpful suggestions by the reviewers. Below is a point-by-point description of made changes to the manuscript as a response to the major comments and questions raised by each reviewer.
* * *
**Comments by reviewer #1**

The authors describe how CCMs are a paleo-permafrost proxy and their importance to mid- and lower-latitude sites. What do the presence of CCMs in this northern-most location tell us about permafrost in this high-latitude site? It's clearly interesting that these CCMs appear to have formed during a short-term extreme event on the ice sheet in 1889. A little more context of why this is an important finding in the Discussion and Conclusion sections would be helpful for the broad audience of this journal.

*We have changed the phrasing in the introduction to make it clearer that $CCC_{fine}$ cannot be used as palaeo-permafrost proxy (LINE 44).*

*We have added more context to the Discussion and Conclusion sections in LINES 370-374 and LINES 393-396, describing how the CCMs can be used to reconstruct the spatial extent of melting conditions in an area outside of the ice sheet and how linking their formation to the summer melt episode of 1889 CE provides hints about the rate of formation.*

The closest ice core record would be from Flade Isblink, and the authors report that this record does not suggest excess melting in 1889, but a recent publication shows that there was a high concentration of black carbon in 1889 at Flade Isblink (Eckhardt et al., 2023). Understandably, the perfect ice core record does not exist to prove the theory that these CCMs were formed due to excess meltwater from the 1889 widespread melting event that occurred at higher elevations on the Greenland Ice Sheet. With the increased concentrations of black carbon at the Flade Isblink site so close to Cove Cave, it does appear that the melting event could have happed in this region too. I suggest including the Flade Isblink black carbon record to bolster this hypothesis.

*We have included the mentioned record/publication in the manuscript (LINES 360-361).*

Specific comments

*We have changed the manuscript according to all of the reviewer's suggestions in this section.*

Figure 1: are the blue lines indicating ice margins referring to present-day ice margins, or ice margins from the former presence of additional ice caps that are mentioned in the last sentence of the figure caption?

*We have changed the figure caption and added "(light blue polygon)" and "(blue lines)" to the respective sentences.*
* * *
**Comments by reviewer #2 – Connor Turvey**

37 – It would be good to explicitly state the criteria for differentiating $CCC_{fine}$ from $CCC_{coarse}$, presumably a grainsize limit.

*We have added a statement on differentiating criteria to the introduction (LINES 36-39).*

70 – It would be better to report an approximate distance from Cove Cave to the weather stations rather than just saying 'closest'.

*The approximate distances are now included in the manuscript (LINES 74-75).*

83 – A quick definition for what 'inactive' means in this context might be helpful, presumably it is common in speleothem geology but I am unfamiliar with it.

*Done (LINE 88).*

91 – Should probably change "rather low" to something less casual.

*Done (LINE 97).*

105 – Were there any obvious visual differences (color, texture etc) between the samples during collection?

*There were not. We have added a sentence to the text (LINE 111).*

110 – Clarity could be improved here, are you taking sample KC19CCC-4, splitting it into different mineral fractions and then mixing the relative amounts? Or are you mixing KC19CCC-4 with another phase?

*We have changed the phrasing to improve clarity, stating that the sample was split into different mineral fractions and the relative amounts were mixed (LINE 116).*

116 – More analytical details for the mineralogy and crystal morphology analysis would be good. For example, with the XRD what was your scan range and analysis time?

*We have added more detail (LINE 121).*

166 – Why could you not ID the very fine brownish crystals? Even if it was too fine to manually separate under a microscope for analysis you had XRD data and could identify the other crystals in the sample so I would have thought it should be possible by process of elimination.

*We have added the most likely composition of the very fine brownish crystals based on the XRD results (LINES 172-173).*

171 – An XRD figure either here or in the appendices showing the results from the 4 samples would be very useful as it would allow for easy comparison between the mineralogy of the four samples, rather than just having it written out.

*We have added X-ray diffractograms to the appendix.*

249 – SEM may provide useful insights here, were any textures observed that could only be explained by synchronous formation (crystals intergrown etc)

*We have added a statement that the examination of SEM images yielded inconclusive results (LINES 263-265).*

250 – Any idea where does the quartz, dolomite and potassium feldspar in KC19CCC-1 come from? Country rock?

*We have added the most likely sources to the manuscript (LINES 259-261).*

280 – No theory as to why your difference between δ18O values of your CCCfine vs. common speleothems is different to that seen in other studies?

*We have added our hypothesis to the text (LINES 294-297), saying that the isotopic composition of the source water differed.*

314 – Would be good to have a reference supporting your claim that there are not CCMs overserved in other caves in the area.

*To the best of our knowledge, there are no other references to support this. We have added to the text that the statement is based on the findings of two expeditions that were part of this project (LINE 331).*
* * *
**Comments by reviewer #3**

I am particularly intrigued by the interpretation of "a few days," specifically at LINES 23-24: "We relate the CCM formation to a combination of black carbon deposition and anomalously high temperatures, which occurred over a few days, in the summer of 1889 CE." The time constraint of "days" is an extraordinarily statement – the fact that a few days of extremely high temperatures caused widespread melting over northeast Greenland is an important finding, if it's true. However, I find the author's reasoning for relating CCM growth to this extreme climate event (a few days of warming) insufficient. There is not a thorough explanation for why authors jump to "days"? In the paper, the only citation is Neff et al. (2014). The authors need to add more explanation to this interpretation.

*We have rephrased LINES 351-357 to make it clearer that the time constraint of "a few days" was not deduced by our samples but is based on a study by Neff et al. (2014). The summer melt episode of 1889 CE itself is recognised in several publications.*

Abstract: I am not sure why authors include the information about CCCfine δ18O values in the abstract? It is my understanding that they do not use this data to make any interpretations?

*We have removed the stable isotope values from the abstract.*

Line 39: Though the authors link CCC formation as a "useful proxy for paleo-permafrost," they do not state clearly whether the formation of CCC=permafrost is present? It may be worth stating this explicitly for readers who are unfamiliar with CCC.

*We have added a clear statement regarding this (LINES 41-42).*

Line 40: Please state the size difference between CCCcoarse and CCCfine. Are CCC samples separated into "coarse" and "fine" categories by eye? By measurement?

*We have added a statement on the differentiating criteria (LINES 36-39).*

Line 45: Is there a reason the authors report the CCM subtypes as $CCC_{coarse}$ and $CCC_{fine}$ versus $CCM_{coarse}$ and $CCM_{fine}$? Lines 37-45 explain the difference between CCC and CCM, but then authors refer their CCM samples as CCC? Please clarify, because right now it seems these two are equivalent.

*We have dropped the term $CCC_{fine}$ when referring to the carbonate fraction of our samples throughout the manuscript.*

Line 48: "recently called into question." How? Please briefly state this. Perhaps move line 50 up to follow this sentence, since I believe it's because of detrital thorium contamination?

*We have included a statement to better explain this (LINES 50-52).*

Figure 4: It is difficult to discern the difference between the light vs. darker gray shading colors. A suggestion to make one of the categories black?

*We have changed the colours in Figure 4.*

Line 125: Please specify where common speleothems were collected. Was it the same cave? Right now it is just reported as "in the study area," which is not enough information. This could help shed light on the δ18O difference between common speleothem and CCCs?

*We have specified where the samples were collected and have added the spatial extend of the study area (LINES 131-132).*

Section 5.2: Are authors interpreting the low δ18O values as reflecting contribution of precipitation from the Arctic air mass? Doesn't this location primarily receive precipitation from the Arctic air mass? Why is this significant? Also, the Greenland common speleothems have a higher δ18O value than CCM δ18O, but they were collected from the same cave? If they are from the same "northeast Greenland" cave, then they should receive precipitation from the same source, and therefore should have the same δ18O? I see at Lines 278-281 the authors address this difference, but do not provide a reason why? Please explain? Even if the authors are not sure why this is, that should be stated. As of now, it is unclear what the assumption of this is, and I find the discussion of the stable isotope data not sufficient.

*We have changed the whole paragraph to focus on the latitudinal shift towards lower $\delta^{18}O$ values that can be observed in cryogenic cave minerals as well as common speleothems (LINES 286-288).*

*We have also added a hypothesis that the isotopic composition of the source water differed between CCMs and common speleothems from Cove Cave (294-297).*

Section 5.2 (continued): What is significant about the Greenland CCMs δ18O overlap with mid-latitude caves? This is not discussed, and I'm a bit confused why this is significant.

*We have added a statement in LINES 289-291 to explain the significance of this finding.*